# Impact of an Antimicrobial Stewardship Program Intervention Associated with the Rapid Identification of Microorganisms by MALDI-TOF and Detection of Resistance Genes in ICU Patients with Gram-Negative Bacteremia

**DOI:** 10.3390/antibiotics11091226

**Published:** 2022-09-09

**Authors:** Aléia Faustina Campos, Tiago Arantes, Amanda Magalhães Vilas Boas Cambiais, Ana Paula Cury, Camila Guimarães Tiroli, Flávia Rossi, Luiz Marcelo Sa Malbouisson, Silvia Figueiredo Costa, Thaís Guimarães

**Affiliations:** 1Department of Infection Control, Instituto Central, Hospital das Clínicas, University of São Paulo, São Paulo 05508-220, Brazil; 2Pharmacy Division, Hospital das Clínicas, University of São Paulo, São Paulo 05508-220, Brazil; 3Microbiology Laboratory, Central Laboratory Division, Hospital das Clínicas, University of São Paulo, São Paulo 05508-220, Brazil; 4Division of Anesthesiology, Hospital das Clínicas, University of São Paulo, São Paulo 05508-220, Brazil; 5Infectious Diseases Department, Hospital das Clínicas, University of São Paulo, São Paulo 05508-220, Brazil

**Keywords:** antimicrobial stewardship, rapid diagnostics tests, gram-negative bacteremia, mortality, days of therapy, antimicrobial consumption

## Abstract

Combination of strategies for rapid diagnostics tests (RDT) with real-time intervention could improve patient outcomes. We aimed to assess the impact on clinical outcomes, antimicrobial consumption, and costs in patients with gram-negative bacteremia. We designed a quasi-experimental study among 216 episodes of gram-negative bacteremia using RDT (MALDI-TOF and detection of resistance genes) directly from blood culture bottles combined with real-time communication of results. Our study did not demonstrate impact on 30-day mortality (25% vs. 35%; *p* = 0.115). Hospital and ICU length of stay were significantly lower in the intervention period ((44 days vs. 39 days; *p* = 0.005) and (17 days vs. 13 days; *p* = 0.033)), respectively. The antimicrobial consumption was 1381 DOT/1000 days in the pre-intervention period compared to 1262 DOT/1000 days in the intervention period (*p* = 0.032). Antimicrobials against gram-positive and carbapenems had a significantly reduced consumption in the intervention period. Our intervention showed no impact on 30 days-mortality, but demonstrated an impact on hospital and ICU length of stay, as well as antimicrobials consumption and costs. Knowledge of resistance genes adds value and information for safe decision making that can result in direct and indirect benefits related to the economic burden of antibiotic overuse and bacterial resistance.

## 1. Introduction

Despite advances in antimicrobial therapy, gram-negative bloodstream infections are still a threat to hospitalized patients. To minimize these threats, early administration of appropriate antimicrobial therapy is critical, as is antimicrobial streamlining to optimal therapy, particularly in a setting of a high prevalence of multidrug-resistant organisms (MDROs) and in low-income countries with scarce therapeutic options. The benefits associated with the de-escalation of antimicrobial therapy include improvements in clinical outcomes, decreased healthcare costs, lower risk of adverse drug events, and less selective pressure for resistance and superinfections [1,2]. Indeed, patients who develop bloodstream infections (BSIs) by MDROs have limited therapeutic options and are at great risk for mortality, complications, and prolonged hospitalization [3].

In 2016, the Infectious Diseases Society of America (IDSA) published the antimicrobial stewardship program (ASP) guidelines. To improve clinical outcomes, in addition to conventional methods for blood specimens, they specifically recommend the use of rapid diagnostic tests (RDTs) with ASP support as well as an intervention. The implementation of RDTs aimed at identifying the causative pathogen in bacteremia may allow for earlier narrowing of antimicrobial therapy [4,5]. 

Several studies have established that the combination of strategies for RDT with real-time intervention by antimicrobial stewardship teams could improve patient outcomes [6,7,8]. However, in a recent meta-analysis, the authors demonstrated that rapid detection of pathogens did not influence mortality [9]. There is very little data in Brazil assessing the impact of stewardship programs on clinical outcomes. 

Therefore, the primary purpose of this study was to assess the impact on clinical outcomes, antimicrobial consumption, and its related cost in patients with gram-negative bacteremia using rapid organism identification through MALDI-TOF and detection of resistance genes directly from blood culture bottles combined with real-time communication of results. The second purpose was to compare the time taken to perform antimicrobial identification (turnaround time, TAT) of pathogens in the pre-intervention and intervention periods.

## 2. Materials and Methods

### 2.1. Study Design and Patient Population

This was a single-center, pre-post intervention, quasi-experimental study conducted in a 925-bed quaternary care teaching facility in Instituto Central, located in São Paulo, Brazil. All adult patients (>18 years old) admitted to the intensive care units (ICUs) with blood culture positive for gram-negative organism between March 2018 to May 2019 (pre-intervention period) and September 2020 to October 2021 (intervention period) were screened for eligibility (Figure 1). Excluded were patients whose index blood culture grew >1 microorganism species (polymicrobial cultures), who died 24 h after treatment, with blood cultures collected only from the central venous catheter, who had a new episode of infection before 14 days of positive blood culture (index), and who had bacteremia not identified from MALDI-TOF. Pre-intervention patients were evaluated via retrospective chart review, while patients in the intervention period were prospectively reviewed as cultures became positive without blinding. 

The design of this study was in accordance with the ethical standards of our local ethics committee and was approved under protocol number 7370; financial resources were granted by FAPESP no.2018/24021-0.

### 2.2. Laboratory Procedures

For both periods, blood cultures were identically analyzed for the presence of microorganisms using the BACTEC FX^TM^ automated blood culture system, which contains culture media with suitable nutritional and environmental conditions for the most common organisms found in the blood. Inoculated bottles are placed into the instrument (BACTEC FX^TM^, (Becton Dickinson Instrument Systems, Sparks, MD, USA), incubated, and continuously monitored for growth. Once an organism was flagged as positive, Gram staining was performed, and results were posted in the patient’s electronic medical record (EMR). A pallet sample from positive blood culture bottles was inoculated on solid media and incubated overnight in 5% CO_2_ at 35 °C. After bacterial growth on these plates, MALDI-TOF MS (VITEK MS, ioMérieux, Rio de Janeiro, Brazil) was performed from the colonies. Antimicrobial susceptibility testing was performed with the VITEK-2 XL system (bioMérieux, MarcyL’Étoile, France) using AST-239 cards (bioMérieux, MarcyL’Étoile, France), E-test (bioMérieux, Durham, NC, USA), disk diffusion (Becton Dickison, Sparks, MD, USA), or broth microdilution according to the Clinical and Laboratory Standards Institute (CLSI) document [10]. In both the pre- and intervention periods, positive blood cultures were evaluated by using the same conventional microbiologic procedures described. However, in the intervention period, the laboratory performed MALDI-TOF MS directly from blood culture bottles after an organism flagged positive and performed genetic resistance determinants as described by Prod‘hom et al. and Galiana et al. [11,12]. 

During the pre-intervention period, the microbiology laboratory results were reported once a day, seven days a week, on the patient’s electronic medical record (EMR) without verbal notification to the ICUs prescribers. Gram stains were routinely performed 24 h a day, every day.

Only during the intervention period were Gram stain results promptly communicated by the microbiology laboratory personnel to the researcher of this study and then directly to a prescriber at the ICUs. Gram stain results received from 5 p.m. to 6 a.m. were communicated to the prescriber in the following morning. In this period, we incorporated detection of resistance genes.

The identification of genetic resistance determinants was performed using the XGEN MULTI SEPSE FLOW CHIP (Mobius) kit for gram-negative bacteria with the HybriSpot equipment (HS12 AUTO). It is a 4-h run time qualitative in vitro test that detects the presence of bacterial nucleic acid in addition to the main antimicrobial resistance genes directly from a positive blood culture bottle without the need for DNA extraction. It consists of a closed platform with commercial kits that uses the real-time PCR methodology for the rapid diagnosis of bacterial infections in clinical species. The XGEN MULTI SEPSE LYO is an RDT with the ability to detect gram-negative bacteria species of interest to our study (*Pseudomonas aeruginosa, Acinetobacter baumannii, Stenotrophomonas maltophilia, Escherichia coli, Klebsiella pneumoniae, Serratia marcescens*), two genera of gram-negative bacteria (Enterobacterales and *Proteus* spp.) and 17 β-lactamase enconding genes: *bla*_SHV_, *bla_CTX-M_*, *bla*_KPC_, *bla*_SME_, *bla*_IMI_, *bla*_GES_, *bla*_VIM_, *bla*_GIM_, *bla*_SPM_, *bla*_NDM_, *bla*_SIM_, *bla*_IMP_, *bla*_OXA23_, *bla*_OXA24_, *bla*_OXA48_, *bla*_OXA51,_ and *bla*_OXA58_.

The MALDI-TOF and the detection of the molecular genes were performed twice daily (morning and afternoon), seven days a week, and all results were communicated on the same day to the researcher of this study. This notification process coming from the laboratory was performed 24 h a day, seven days a week throughout the intervention period only. The antimicrobial stewardship team for this study was composed of two infectious diseases physicians. One of them who received a real-time notification of ICUs patients with positive blood cultures containing gram-negative bacteria rapidly provided the information to prescribers. Then, antimicrobial therapy was managed according to the blood culture results and evidence-based antibiotic recommendations available within institutional guideline [13] and international literature [14,15]. Empirical therapy was defined as antibiotic administration before pathogen identification. Carbapenem resistance was defined as gram-negative microorganisms resistant to imipenem and/or meropenem.

### 2.3. Data Collection

We collected data on demographic characteristics, clinical conditions at admission, microbiology data, antibiotic therapy, the severity of illness based on SAPS 3 at ICU admission and PITT score on the day of bacteremia, reason of hospital admission, and antimicrobial empiric therapy prescribed. Infection-related characteristics collected included source, causative pathogen, and susceptibility data. The source of bacteremia was determined according to the definitions published by the Centers for Disease Control and Prevention (CDC) [16]. Data from antibiotics used was described as days of therapy (DOT) [4,17]. 

### 2.4. Outcomes

The outcomes evaluated included 30-day all-cause mortality, hospital and ICUs length of stay after blood culture positivity, time to identification of Gram stain, MALDI-TOF, PCR and antibiotic susceptibility test (TSA) from bottles following blood culture positivity (turnaround time, TAT), antimicrobial consumption (DOT/1000 days present), and costs. 

The length of ICU and hospital stay was measured from positive blood culture date until death or ICU/hospital discharge. 

Antimicrobial days of therapy (DOT) were collected for all antibacterial agents administered during the maximum twenty-one inpatients days post-culture collection or end of treatment. It was expressed as DOT per 1000 days present. DOT per 1000 days present was calculated as the sum of the days on therapy for all systemic antibiotics, normalized per 1000 days present. 

Antimicrobial costs were calculated based on drug acquisition costs by the pharmacy department for each drug. We calculated the costs in local currency, but for the analysis, we made the conversion to United States dollar (quotation of 12 May 2022).

### 2.5. Statistical Analysis

We calculated the sample size considering an absolute difference of 36% between the groups regarding the mortality outcome as described by Timbrook (6) and considering a statistical power of 80% and a statistical significance of 5%, and we found a need for 44 patients in each group.

Descriptive statistics were performed for all continuous (median, IQR) and categorical (number, percent) data. Continuous variables were compared using the Student’s *t*-test or the Mann–Whitney U test where appropriate, and categorical variables were compared using Pearson’s chi-square test and Fisher’s exact test, if appropriate. Univariate and multivariate analyses were performed with SPSS for Windows, version 27.0 (SPSS Inc., Chicago, IL, USA), and a two-tailed *p* value of 0.05 was considered statistically significant.

To identify independently associated with mortality at 30-day, a multivariate forward, stepwise logistic regression analysis was performed with *p* < 0.05 to report. This was preceded by conducting univariate analysis to determine variables to be included in the multivariable model by Cox regression. Odds ratio (OR) with 95% confidence interval (CI) were calculated.

## 3. Results

Adult patients admitted in an ICU setting from March 2018 to May 2019 (pre-intervention period) with a positive blood culture of gram-negative bacteria identified through Gram stain were included and compared to patients with the same conditions during the period of September 2020 to October 2021 (intervention period) (Figure 1). A total of 346 episodes of bacteremia were evaluated for inclusion. After the defined inclusion criteria, 216 episodes of bacteremia from 213 patients were included in the final analysis: 114 bacteremia from 112 patients in the pre-intervention period and 102 bacteremia from 101 patients in the intervention period. 

The groups analyzed in the two periods were similar, except for the severity where we could note a larger SAPS3 in the intervention period, as well as the presence of BSI secondary to pulmonary focus was more prevalent in the intervention period. The median age of patients in the pre-intervention and intervention periods was 56 years (41–62) and 59 years (47–69), *p* = 0.057, respectively, and males were more prevalent in both periods. Most isolates in the pre-intervention period were primary BSIs (50.8%), followed by abdominal (14.9%) and respiratory (11.4%) sources. In the intervention period, the source of BSIs were primary (47.1%), followed by respiratory (27.5%) and urinary (8.8%). Pathogen prevalence was similar in both periods, with *K. pneumoniae* being the most prevalent bacteria (43% vs. 32.4%, *p* = 0.043). The distribution of microbiological isolates was similar in both periods, except for the prevalence of *P. aeruginosa* where we noticed a significant difference in the second period (6.1% vs. 21.6%; *p* = 0.002). In addition, the rate of carbapenem-resistant Enterobacterales in both periods was similar, 24.6% vs. 15.7%, *p* = 0.14, respectively. In the intervention period, the CTX-M encoding gene was the most common resistance gene detected (68.6%), followed by OXA (41.2%) and KPC (27.4%) encoding genes. There were no VIM or IMP-encoding genes detected during the study period and only one metallo-β-lactamase was identified (an NDM-encoding gene in *K. pneumoniae*). We also found a statistical difference during the intervention period when we compared the empirical antimicrobial regimens based on three drugs. We evaluated the mean duration of antimicrobial therapy that was significantly different (9.13 vs. 7.8 days; *p* = 0.013) between periods; however, the median of both periods was identical (8 days). Table 1 demonstrates patients’ demographic characteristics.

Our study did not demonstrate impact-related 30-day mortality between periods. Hospital length of stay was significantly lower in the intervention period (44 days vs. 39 days; *p* = 0.005) as the ICU length of stay (17 days vs. 13 days; *p* = 0.033). The clinical outcomes are summarized in Table 2.

The multivariate analysis of 30-day mortality identified only SARS-CoV-2 as an independent predictor as presented in Table 3.

The median of antimicrobial consumption over the 21 days post-culture collection was 1.381 DOT/1000 days present (IQR 1.103–2.251) in the pre-intervention period compared to 1.262 DOT/1000 days present (IQR 1.063–1.662) in the intervention period (*p* = 0.032) as shown in Table 4. When analyzing antibiotic consumption according to gram-negative or gram-positive coverage, only the agents against gram-positive bacteria had a significant reduced consumption in the intervention period, 475 (238–761) vs. 270 (139–467), *p* = 0.004, respectively. Despite having performed an analysis of all gram-negative agents, no difference was noted between the periods (*p* = 0.067), but there was a statistical difference for the specific consumption of carbapenem drugs between periods (*p* = 0.04). In addition, considering only the direct antimicrobial costs for all drugs, a lower cost was noted for gram-positive bacteria coverage drugs and for carbapenem drugs in the intervention group.

The median time to positivity (TTP) for the index blood culture was 11 h 53 min (9 h 47 min–17 h 26 min) vs. 12 h 29 min (10 h 07 min–18 h 18 min) and not significant between periods (*p* = 0.302), as shown in Table 5. However, in the second period the performance of MALDI-TOF directly from blood culture significantly reduced the median time of identification from 26 h 31 min (20 h 33 min–33 h 42 min) to 9 h 31 min (6 h 28 min–15 h 10 min) with a statistically significant difference. In addition, the TAT between TTP and Gram stain and AST were lower in the intervention group. Figure 2 demonstrate TAT between periods.

## 4. Discussion 

We conducted a pre- and post-intervention study to evaluate its influence on clinical and economic outcomes. Our initial objective was to verify whether the implementation of MALDI-TOF in our hospital had brought benefits in terms of clinical outcomes. Thus, we started collecting data without intervention. During this period, we observed that the microbiology laboratory, despite having the MALDI-TOF, waited the 24-h incubation period for bacteria identification. Therefore, after identifying the species directly from bottles of positive blood culture, we started to perform this technique during the intervention period, in addition to a rapid communication to the prescribing physician and the detection of resistance genes [11,12,18].

In our study, rapid organism identification and susceptibility reporting with real-time antimicrobial stewardship efforts were not associated with a significant reduction in all-cause 30-day mortality, but it was statistically significant for the hospital and ICU LOS and for direct antimicrobial costs.

Perez and colleagues [19] conducted a pre-post quasi-experimental study integrating MALDI-TOF organism identification plus antimicrobial stewardship in patients with gram-negative bacteremia. Like our study, they demonstrated a non-significant reduction in mortality (10.7% vs. 5.6%, *p* = 0.19). However, they demonstrated statistically significant reductions in length of hospitalization (11.9 vs. 9.3 days, *p* = 0.01) and total hospital costs ($45,709 vs $26,126, *p* = 0.009). Mortality benefits may be difficult in pre- and post-intervention studies where confounding factors cannot been controlled. In a systematic review and meta-analysis of 31 studies, RDT was associated with a decreased mortality risk and LOS, as well as improved time to effective therapy, compared with conventional microbiologic methods. However, they found that mortality risk decreased significantly with RDT in the presence of antimicrobial stewardship, (OR, 0.64; 95% CI (0.51–0.79)) with ASP vs. (OR, 0.72; 95% CI (0.46–1.12)) without ASP [4]. In a recent review published by the Cochrane Library, including six trials with 1638 participants, the results showed that the benefits of RDT have not demonstrated to improve mortality [9].

Despite the benefit of reduced mortality being the most important outcome, unfortunately our study was also unable to demonstrate such benefit, most likely due to the severity of patients hospitalized in intensive care units where the aggravating bloodstream infection is certainly a risk factor for mortality. The first case of COVID-19 in Brazil was detected in March 2020, in the city of São Paulo, and our hospital became a reference for the care of COVID-19 with all beds destined for COVID-19. Then, the study was interrupted, only returning from September 2020. During the intervention period, we experienced the appearance of SARS-CoV-2 in our ICUs [20,21], which certainly contributed to the greater severity of patients (see SAPS 3) and did not influence the reduction of mortality. Univariate analysis also showed a higher prevalence of pulmonary bloodstream and *P. aeruginosa* as the most prevalent agent in the intervention period. It is correlated with the scenario of higher prevalence of SARS-CoV-2 where the lung is the target organ and can result in secondary infections. Indeed, *P. aeruginosa* showed more prevalent in the intervention period since bacteremia of pulmonary focus is also associated with SARS-CoV-2, as described by Rouzé et al. [22]. We pointed out that during this period, no outbreaks by this agent were identified and both did not remain independent risk factors in the multivariate analysis. However, when the other outcomes of hospital and ICU length of stay were analyzed, the intervention was beneficial.

Regarding the isolated microorganisms, although *K. pneumoniae* and *A. baumannii* sp. showed stability in two periods, we noted an increase in the prevalence of *P. aeruginosa* in the intervention period. This fact can be explained by the COVID-19 pandemic, where several studies have also shown an increase in BSI by *P. aeruginosa* [23]. On the other hand, carbapenem resistance was not statistically different in both periods, both for Enterobacterales and for non-fermenting bacilli.

The resistance rate for Enterobacterales represents a real challenge for empiric treatment decisions in ICUs since more than one-third of gram-negative bacteria were found to be resistant to the carbapenem (20.37%, *n* = 44/216). Brazilian data show that carbapenem-resistant *K. pneumoniae* is the most prevalent pathogen causing BSIs in adults in ICUs [23]. Therefore, empiric combination antibiotic therapy is a reality.

As reported in a Brazilian point prevalence survey, results coming from five ICUs showed a high prevalence of antimicrobial use highlighting the high proportion of combination therapy (51%) to deal with the high rate of MDROs in this setting [24]. Our results from rapid molecular tests performed on blood cultures flagged positive during the intervention period showed a high prevalence of microbiological isolates producers of ESBL (68.6%), oxacilinases (41.2%), and KPC (27.4%). Knowledge of the resistance mechanism before the susceptibility test is a very useful point-of-care and highly recommended tool that certainly contributes to the de-escalation and adjustment of the antimicrobial regimen [3,18,19,25]. It remains to be seen what the long-term impact will be of reducing bacterial resistance.

Considering the economic outcomes, we noted a general reduction in antimicrobial consumption. This was due to the reduction of antimicrobial consumption of gram positives, obviously, because our intervention significantly influenced the discontinuation of unnecessary antimicrobials related to vancomycin. An explanation for this is because a prompt pathogen communication and discussion of streamlining therapy certainly improve treatment considering real-time results that enhance the ICU physician’s ability to adjust antimicrobial therapy sooner.

We chose to analyze the consumption and costs of carbapenems, polymyxins and ceftazidime–avibactam because these antimicrobials are the most empirically used in our multidrug resistance scenario, and we found a significant reduction in the consumption of carbapenems. This can be explained by the higher prevalence of susceptible *P. aeruginosa*. Although this microorganism was more prevalent, there was no change in susceptibility profile since only 3.7% of *P. aeruginosa* isolates were resistant to carbapenems.

We observed an increase in the total cost of antimicrobials. There was an increase in the costs of antimicrobials for gram negatives and in the specific cost of ceftazidime–avibactam. Cost reduction was noted for gram positives, carbapenems, and polymyxins antimicrobials. Our total cost analysis was not lower during the intervention period due to the introduction and standardization of ceftazidime-avibactam in 2020, a novel beta-lactamase inhibitor used for the treatment of carbapenem-resistant Enterobacterales that is a very expensive drug in our country and was not available in the pre-intervention period. The polymyxin class showed no reduction in consumption, but demonstrated a reduction in the cost resulting from the use of polymyxin B in the intervention period, which is a cheaper molecule than colistin.

Decreasing the turnaround time (TAT) of microbiological results seems to improve outcomes, especially in those related to antimicrobial use, as it could promote a faster escalation or de-escalation of the antibiotic treatment [6,8,19]. In our study, the median TAT of the MALDI-TOF results, performed directly from positive-flagged blood cultures, was one day quicker compared with the pre-intervention study group (26 h 31 min vs. 9 h 31 min, *p* < 0.001). Our study also reinforces that performing MALDI-TOF directly from positive blood cultures is safe and reduces TAT. Considering that many microbiology laboratories have the MALDI-TOF MS methodology, the procedure described in our study to carry out the identification of microorganisms directly from the positive blood culture sample must be routinely performed and incorporated in order to have a positive impact on decisions therapeutics. We also demonstrated a reduction in the median time of bacterial detection and reporting of antimicrobial susceptibility (AST) results in the intervention period (54 h 14 min vs. 48 h 28 min, *p* = 0.005) most likely due to the intervention of the researcher in the previous stages of bacterial identification. Munson et al. demonstrated that the Gram stain results reported by telephone to the clinician had an even greater influence on the antibiotic regimen than antimicrobial susceptibility testing [26]. This issue reinforces the importance of effective and real-time communication of positive blood culture from Gram staining, which is a simple, inexpensive method, available in any microbiology laboratory. Although the incorporation of the detection of resistance genes is a costly strategy, it is currently necessary for decision-making. In the scenario of multidrug resistance due to the epidemiology of MDROs, detection of resistance genes in direct multiplex PCR may advocate the investment [27,28].

Our study has some limitations. It was a quasi-experimental study that utilized a convenience sample and therefore lacked randomization. The best methodology to evaluate the effectiveness of health interventions at a population level is interrupted time series studies, which are widely used to evaluate therapeutic measures. However, as we were surprised by the COVID-19 pandemic, it might have contributed to a heterogeneity of the sample in the intervention period where patients were more severe due to SARS-CoV-2 infection, so that the multivariate analysis showed it as the only predictor of mortality. This may have been an important factor for not having found lower mortality in the intervention period. 

We also did not measure the impact of RDT on the time it took to change empiric therapy between periods as this is already widely covered in the literature. As an intervention plan, we introduced, concurrently, the MALDI-TOF and genes detection directly from blood cultures plus antimicrobial stewardship communication, thus making it difficult to delineate the impact of each individual intervention. As pointed out by Doern, there is a confounding role of active antimicrobial stewardship intervention in evaluating outcomes after the implementation of RDTs, as using a paired intervention limits the ability to determine the degree to which each component individually contributes to clinical outcomes [29]. Another limitation was that we did not correlate genotypic results with phenotypic results of isolated microorganisms, and we did not calculate the indirect costs related to the shorter length of hospital stay.

The main strength of our study is to show that a same-day strategy for the communication of MALDI-TOF and molecular resistance genes pathogen results directly from positive blood cultures bottles to the prescriber’s physician at ICUs would represent another opportunity to increase the appropriateness of empirical antibiotic therapy, even though in a setting with a high prevalence of MDROs.

We pointed out that our intervention showed no impact on 30 days-mortality, but demonstrated an impact on hospital and ICU length of stay, as well as antimicrobials consumption and costs. We did demonstrate that TAT could be improved when using MALDI-TOF directly from positive-flagged blood cultures combined with a strategy of stewardship program. Incorporating knowledge of resistance genes before susceptibility testing adds value and information for safe decision making that can result in direct and indirect benefits related to the economic burden of antibiotic overuse and bacterial resistance.

More studies are needed with a larger sample size, longer evaluation time and comparing similar populations without COVID-19 bias to better analyze the mortality outcome.

## Figures and Tables

**Figure 1 antibiotics-11-01226-f001:**
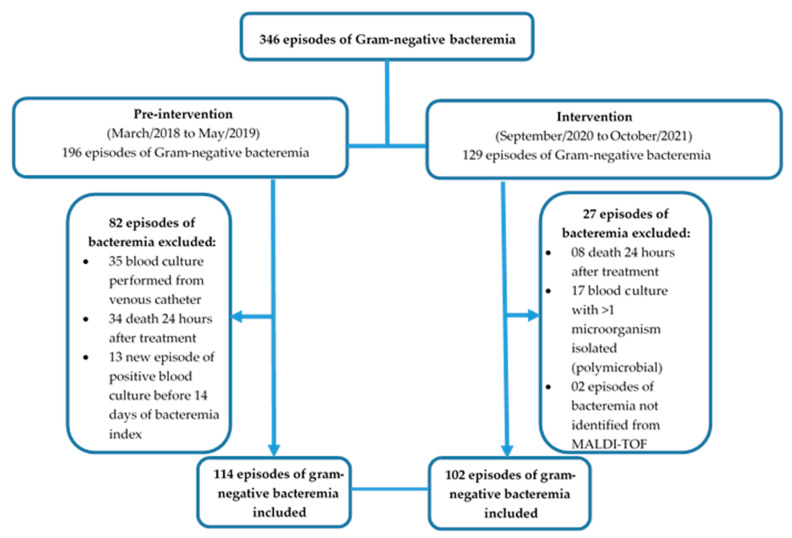
Patient selection flowchart.

**Figure 2 antibiotics-11-01226-f002:**
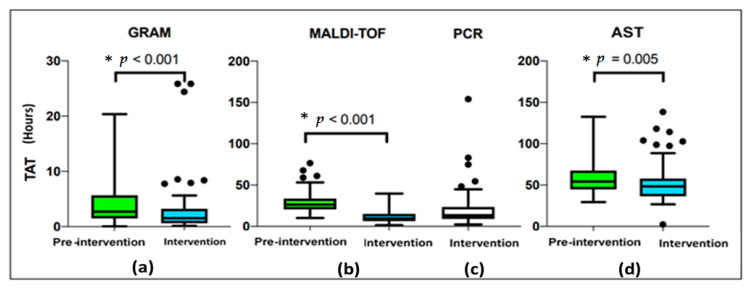
TAT: Turnaround time between pre-intervention and intervention period. (**a**): GRAM: time from bacteria detection until Gram stain. (**b**): MALDI-TOF: time from bacteria detection until MALDI-TOF. (**c**): PCR: time from bacteria detection until PCR for the intervention group only. (**d**): AST: time from bacteria detection until antimicrobial susceptibility test. *: Mann–Whitney test.

**Table 1 antibiotics-11-01226-t001:** Baseline patient characteristics, resistance genes detection, and empirical antimicrobial therapy of patients.

Variable	Pre-Intervention (N = 114)	Intervention (N = 102)	*p*-Value
Median age (range in years)	56 (41–62)	59 (47–69)	0.057 ^a^
Males (%)	78 (68.4%)	65 (63.7%)	0.466 ^b^
End stage renal diseases requiring dialysis, N (%)	37 (32.5%)	42 (41.2%)	0.184 ^b^
Central venous catheter, N (%)	81 (71.1%)	84 (82.4%)	0.051 ^b^
SAPS3 (median, IQR)	56 (45–68)	64 (50–76)	0.005 ^a^
PITT bacteremia score ≥ 6, N (%)	60 (52.6%)	62 (60.8%)	0.228 ^b^
ICU admission, N (%)			
Severe acute respiratory syndrome SARS-CoV-2	0 (0%)	45 (44.1%)	<0.0001 ^b^
Gastrointestinal diseases	25 (21.9%)	5 (4.9%)	0.0003 ^b^
Infectious diseases	26 (21.9%)	8 (7.8%)	0.002 ^b^
Cerebrovascular diseases	16 (14%)	10 (10%)	0.34 ^b^
Trauma	13 (11.4%)	10 (10%)	0.7 ^b^
Kidney diseases	8 (7%)	5 (5%)	0.51 ^b^
Cardiovascular diseases	5 (4.4%)	5 (5%)	0.85 ^b^
Metabolic diseases	3 (2.6%)	1 (1%)	0.36 ^b^
Others	19 (716.6%)	13 (13%)	0.33 ^b^
Blood stream infection source, N (%)			
Primary	58 (50.8%)	48 (47.1%)	0.671 ^b^
Pulmonary	13 (11.4%)	28 (27.5%)	0.005 ^b^
Urinary	9 (7.9%)	9 (8.8%)	1.0 ^b^
Intra-abdominal	17 (14.9%)	7 (6.9%)	0.096 ^b^
Skin and soft tissue	7 (6.1%)	8 (7.8%)	0.823 ^b^
Other sites	10 (8.7%)	2 (2.0%)	0.060 ^b^
N (%) of organisms isolated			
*K. pneumoniae*	49 (43.0%)	33 (32.4%)	0.043 ^b^
*E. coli*	19 (16.6%)	13 (12.7%)	0.536 ^b^
*A. baumannii*	9 (7.8%)	17 (16.7%)	0.077 ^b^
*P. aeruginosa*	7 (6.1%)	22 (21.6%)	0.002 ^b^
*Enterobacter* spp.	5 (13.1%)	5 (4.9%)	0.064 ^b^
Others Enterobacterales	13 (11.4%)	8 (7.8%)	0.515 ^b^
Others non fermentative	2 (1.7%)	4 (3.9%)	0.580 ^b^
N (%) Carbapenem-resistance (N = 76)			
Enterobacterales	28 (24.6%)	16 (15.7%)	0.141 ^b^
*P. aeruginosa*	3 (2.7%)	5 (5.0%)	0.602 ^b^
*A. baumannii*	8 (7.0%)	16 (15.7%)	0.071 ^b^
N (%) Genes Resistance (N = 51)			
*bla* _CTX-M_	-	30 (68.6%)	
*K. pneumoniae*	-	22	
*E. coli*	-	4	
*Enterobacter* spp.	-	2	
*P. aeruginosa*	-	1	
*Burkholderia cepacia complex*	-	1	
*bla* _KPC_		14 (27.4%)	
*K. pneumoniae*	-	13	
*E. coli*	-	1	
*bla*_OXA-23_, *bla*_OXA-24_, *bla*_OXA-51_		21 (41.2%)	
*A. baumannii*	-	20	
*P. aeruginosa*	-	1	
SHV-enzymes		5	
*K. pneumoniae*	-	5 (9.8%)	
*bla*NDM-		1	
*K. pneumoniae*	-	1 (2.0%)	
Empirical Antimicrobial regimen			
Monotherapy	56 (49.1%)	50 (49.0%)	0.988 ^b^
2 antibiotics	38 (33.3%)	48 (47.1%)	0.040 ^b^
3 antibiotics	20 (17.5%)	4 (3.9%)	0.001 ^b^
Duration of antimicrobial therapy (mean; days)	9.13	7.8	0.013 ^a^

Data presented as N (%) or median, interquartile range (IQR). a = Mann–Whitney, b = Pearson Chi-Square, SAPS3—simplified acute physiology score 3, ICU—intensive care unit, CTX-M—Cefotaximase, KPC—*K. pneumoniae* carbapenemase, OXA—oxacilinases, SHV—Sulfhydryl Variable, NDM—New Delhi metallo-β-lactamase.

**Table 2 antibiotics-11-01226-t002:** Clinical outcomes of patients included in the study.

Characteristic Median, IQR	Pre-Intervention N = 114	Intervention N = 102	*p*-Value
30-day mortality	29 (25%)	36 (35%)	0.115 ^b^
Hospital LOS	44 (20–59)	39 (14–48)	0.005 ^a^
ICU LOS	17 (7–22)	13 (5–16)	0.033 ^a^

a = Mann–Whitney, b = Pearson Chi-Square, ICU—Intensive care unit, LOS—Length of stay.

**Table 3 antibiotics-11-01226-t003:** Univariate and multivariate analysis of the risk factors of 30-day mortality in patients included in the study.

	Univariate Analysis			
Variable	Death			
	No	Yes	HR (CI 95%)	*p*-Value
	(N = 151)	(N = 65)		
Pre intervention, N	49 (43%)	65 (57%)		
Intervention, N	34 (33.3%)	68 (66.6%)	1.38 (0.97–1.95)	0.072
SARS-CoV-2, N	9 (20%)	36 (80%)	1.74 (1.17–2.59)	0.006
SAPS3 (median, IQR)	51 (45–67)	62 (50–74)	1.01 (1.00–1.02)	0.083
Gastrointestinal diseases, N	17 (56.6%)	13 (43.3%)	1.18 (0.85–1.64)	0.333
Infectious diseases, N	12 (14.5%)	22 (16.5%)	0.83 (0.52–1.32)	0.432
Blood stream infection source: pulmonary, N	18 (43.9%)	23 (56.1%)	1.39 (1.07–1.80)	0.036
Organism isolated: *P. aeruginosa*	11 (37.9%)	18 (62.1%)	1.38 (1.09–1.74)	0.002
	Multivariate analysis			
	OR	CI95%		*p*-value
SARS-CoV-2	1.54	(1.02–2.29)		0.036
Blood stream infection source: pulmonary	1.75	(0.87–3.51)		0.113
Organism isolated: *P. aeruginosa*	1.26	(0.56–2.83)		0.573

HR: Hazard Ratio; OR: Odds Ratio; CI: confidence intervals.

**Table 4 antibiotics-11-01226-t004:** Antimicrobial consumption in DOT per 1000 days present and costs following culture collection (21 days).

Antimicrobial Consumption and Cost	Pre-Intervention	Intervention	*p*-Value	Percent of Change
Consumption (DOT/1000 days present, median, IQR)				
All antimicrobials	N = 114 1.381 (1.103–2.251)	N = 102 1.262 (1.063–1.662)	0.032 ^a^	
Antimicrobial for gram-negative bacteria	N = 114 1.281 (1.004–1.775)	N = 102 1.172 (1.006–1.427)	0.067 ^a^	
Antimicrobial for gram-positive bacteria	N = 52 475 (238–761)	N = 43 270 (139–467)	0.004 ^a^	
Carbapenems	N = 76 836 (504–1056)	N = 72 543 (301–991)	0.040 ^a^	
Colistin/Polymyxin B	N = 45 722 (390–932)	N = 40 881 (462–1001)	0.299 ^a^	
Ceftazidime/avibactam	N = 3 931 (725–1022)	N = 8 911 (823–940)	0.865 ^a^	
Costs (Sum USD)				
All antimicrobials	N = 114 $23,937.60	N = 102 $31,126.05	0.30	+30%
Gram-negative antimicrobials	N = 114 $23,331.30	N = 102 $30,332.39	0.283	+33%
Gram-positive antimicrobials	N = 52 $627.00	N = 43 $202.68	0.008	−78%
Carbapenems	N = 76 $6003.20	N = 72 $3938.63	0.039	−34%
Colistin/Polymyxin B	N = 45 $5450.26	N = 40 $3030.88	0.067	−47%
Ceftazidime/avibactam	N = 3 $4471.50	N = 8 $19,949.77	0.008	+346%

Antimicrobial for gram-negative bacteria coverage: carbapenem, colistin/polymyxin B, ciprofloxacin/levofloxacin, amikacin, ceftriaxone, piperacillin/tazobactam, tigecycline, ceftazidime, cefepime, ceftazidime/avibactam, gentamycin. Antimicrobial for gram-positive bacteria coverage: oxacillin, vancomycin, and linezolid. Carbapenem: imipenem, meropenem and ertapenem. a: Mann-Whitney test, IQR—interquartile range, DOT—Days of therapy. USD = United States dollars.

**Table 5 antibiotics-11-01226-t005:** Turnaround time (TAT) comparison between pre-intervention and intervention period.

Turnaround Time (TAT) (Hours, Minutes)	Pre-Intervention N = 114	Intervention N = 102	*p*-Value
Time to positivity (TTP), median, (IQR)	11 h 53 min (9 h 47 min–17 h 26 min)	12 h 29 min (10 h 07 min–18 h 18 mim)	0.302 ^a^
Time between TTP and Gram stain, median, (IQR)	2 h 43 min (1 h 30 min–5 h 40 min)	01 h 32 min (37 min–3 h 11 min)	<0.001 ^a^
Time between TTP and MALDI-TOF, median, (IQR)	26 h 31 min (20 h 33 min–33 h 42 min)	9 h 31 min (6 h 28 min–15 h 10 min)	<0.001 ^a^
Time between TTP and PCR, median, (IQR)		8 h 49 min (7 h 57 min–13 h 33 min)	
Time between TTP and AST, median, (IQR)	54 h 14 min (44 h 49 min–67 h 12 min)	48 h 28 min (36 h 33 min–57 h 47 min)	0.005 ^a^

a: Mann–Whitney test, IQR—interquartile range, MALDI-TOF—matrix-assisted laser desorption ionization—Time of Flight matrix assay, TAT—turnaround time, TTP—time to positivity, AST—antimicrobial susceptibility test.

## Data Availability

Not applicable.

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
