# Peer review of "Impact of an Antimicrobial Stewardship Program Intervention Associated with the Rapid Identification of Microorganisms by MALDI-TOF and Detection of Resistance Genes in ICU Patients with Gram-Negative Bacteremia"

_antibiotics, 2022, doi:10.3390/antibiotics11091226_

Round 1

Reviewer 1 Report

The authors set out to study the impact of introducing rapid pathogen identification and AST data on clinical outcomes, antibiotic consumption and costs in patients with gram-negative bacteremia. They report that while there was no significant difference in 30-day mortality among patient groups, the patients in the "intervention group" had shorter hospital stays and, predictably, a significantly reduced use in antibiotics that target gram-positive bacteria. Studies such as this are important in highlighting the significance of timely and appropriate antibiotic interventions to resolve infections. However, there are significant limitations in this study. 

1. As the authors noted, the study was conducted on patients from 2020 to October 2021, which coincided with the COVID-19 pandemic. This could have influenced the mortality rate of this patient group compared to the pre-intervention period (from March 2018 to March 2019). The mortality rate among the intervention group was very high (35%) perhaps due to the pandemic. 

2. The patient population used for the study is too small to draw any conclusions, with just over 100 patients in either group.

Reviewer 2 Report

Review on: „Title: Impact of an antimicrobial stewardship program intervention associated with the rapid identification of microorganisms by MALDI-TOF and detection of resistance genes in ICU patients with gram-negative bacteremia“ by Faustina Campos et al. [antibiotics-1864376]

General Comments:

The manuscript by Faustina Campos et al. is well structured and written in an understandable way. The scientific question is clearly answered by the methods used. Furthermore, the topic is highly relevant and confirms results of other research groups using an alternative experimental setting. However, the authors should still consider whether it is at all possible to influence mortality with the help of timely microbiological laboratory diagnostics and, if so, which setting they would suggest for future studies.

Since many microbiology laboratories now have a MALDI TOF MS, the procedure described here for the identification of bacterial species can also be widely adopted. The array platform used here is able to recognise most of the resistance genes that are relevant in practice. Therefore, the procedure described here can be implemented in many microbiological laboratories. For this reason, both the topic and the method described are very relevant for diagnostic users. The methods are clearly described and are completely sufficient to answer the scientific question posed here.

In particular, the use of the MALDI TOF MS and the consideration of a pre- and post-interventional study section should be mentioned here.

Reviewer 3 Report

Campos et al. evaluated the impact of an antimicrobial stewardship in combination with rapid microbiological identification and detection of resistance determinants using a multiplex PCR-based platform on intensive care patients with gram-negative bacteremia. In this study, authors evaluated the impact of the intervention on clinical outcomes (mortality, length of ICU and hospital stay), antimicrobial consumption and cost of antimicrobials using a before and after quasi-experimental design. Authors found a statistically nonsignificant decrease in 30-day mortality, statistically significant shorter hospital and ICU stays, as well as a decreased antimicrobial consumption. There is only a limited number of publications evaluating the impact of antimicrobial stewardship programs in Latin America, thus this research is of interest to the scientific community. Although there are some errors, the manuscript is written well. Nevertheless, there are some major issues which need to be addressed:

a)       The terms intervention-period and post-intervention-period are used as synonyms, which is very confusing. To make it even worse, the definition of the periods is different in the text compared to figure 1. In the text the pre-intervention period was established from march 2018 to march 2019, in figure 1, is from march 2018 to may 2019 and so on…

b)      Quasi-experimental studies using a before- and after-design are problematic and interrupted time series should be preferred when ever possible.

c)       Some definitions are lacking, i.e., carbapenem-resistance

A few additional errors:

L 94: In both, the pre- […] blood cultures were evaluated […]

L 97/108: genetic resistance determinants

L 118: “Enterobacteriaceae and Proteus spp.” or Enterobacterales; “17 β-lactamase encoding genes”

L 203-206: There is kind of a mix up of clinical conditions (bacteremia) and anatomical sites (abdominal and respiratory tract. It should be either anatomical sites (blood stream, abdominal and respiratory tract), samples (blood, bile, surgical samples, […], sputum, […]) or clinical conditions (patients with BSI, intraabdominal infections, respiratory infections). “Most isolates […] were obtained from patients with primary BSIs […]”

L 207: I would suggest using “microbiological isolates” instead of pathogens

L 211-214: Do not use terms like “CTX-M gene”, “VIM or IMP resistance genes”. Instead use “CTX-M-encoding gene, KPC-encoding gene” or “blaCTX-M, blaKPC”; There are many different OXA-enzymes with very different enzymatic activity (something similar applies to SHV-enzymes; table 1).

L 220ff: Table 1 needs to be improved; some example are “p-value”, instead of “P-value”; “Median age (range)” instead of “age, years”; Males (%),…; K. pneumoniae, E. coli, A. baumannii, Enterobacter spp., Carbapenem-resistance: Enterobacterales […]

Round 2

Reviewer 1 Report

I have no further concerns.

Author Response

Dear reviewer,

We are very pleased with your comments.

Reviewer 3 Report

As mentioned previously, quasi-experimental studys with a before and after design are problematic. Although quasi-experimental studys sometimes are the only way to evaluate certain interventions, interupted time series are recommended. There are several other factors which might have had an impact on the study (the COVID-19 pandemic, changes in local antimicrobial resistance patterns, hospital outbrakes, changes in behaviour of prescribers....)

I am sure the investigators invested a lot of energy and made a lot of efforts to complete this study; however, there is too much uncertainty if the results can be attributed to the intervention.

Some additional comments:

L 148: “The length of ICU and hospital stay was measured from positive blood culture date 148 until death or ICU/hospital discharge.“ This is somwhow problematic, since patients dying quickly would actually lower the average LOS.

Again, there are discrepancies between the study period mentioned in the text and the period mentioned in figure 1.

L 131: Why was carbapenem resistance defined as resistance to both imipenem and meropenem?

L 194: Figure 1 – you do not exclude bacteremia; you can exclude a patient with bacteremia or an episode of bacteremia in a patient.

L 211: K. pneumoniae instead of Klebsiella pneumonia (also applies for L 312, L 321....)

L 216-219: Do not use terms like “CTX-M gene”, “VIM or IMP resistance genes”. Instead use “CTX-M-encoding gene, KPC-encoding gene” or “blaCTX-MblaKPC”; There are many different OXA-enzymes with very different enzymatic activity (something similar applies to SHV-enzymes; table 1).

L 218: encondinggenes

L 226: Table 1.  p-value” instead of p-value; “Median age” instead of “Age, median age”; use complete bacterial names only once, afterwards the genus can be abbreviated. “blaKPC, blaOXA-23,…” instead of “KPC-encoding gene, ….” probably would be more elegant

L 275: “Our initial objective was to verify whether the implementation 275 of MALDI-TOF in our hospital had brought benefits in terms of clinical outcomes”. In order to verify if the implementation of MALDI-TOF had brought benefits, the authros should have collected data before the introduction of MALDI-TOF to be able to compare outcomes.

Author Response

Dear reviewer,

See attached.

Round 3

Reviewer 3 Report

As mentioned in previous reviews, there are some serious limitations of the study, thus the results can not be attributed to the intervention. I am aware of the author's efforts and the lack of studies regarding antimicrobial stewardship in Latin America. Nevertheless, personally I don’t agree with the publication of the study.